# Disruption of Bacterial Thiol-Dependent Redox Homeostasis by Magnolol and Honokiol as an Antibacterial Strategy

**DOI:** 10.3390/antiox12061180

**Published:** 2023-05-30

**Authors:** Yanfang Ouyang, Xuewen Tang, Ying Zhao, Xin Zuo, Xiaoyuan Ren, Jun Wang, Lili Zou, Jun Lu

**Affiliations:** 1Engineering Research Center of Coptis Development and Utilization/Key Laboratory of Luminescence Analysis and Molecular Sensing, Ministry of Education (Southwest University), College of Pharmaceutical Sciences, Southwest University, Chongqing 400715, China; ouyangyanfang@163.com (Y.O.); 18832819339@163.com (X.T.); zhaoying_totoro@163.com (Y.Z.); 15111855313@163.com (X.Z.); 2Division of Biochemistry, Department of Medical Biochemistry and Biophysics, Karolinska Institutet, SE-171 77 Stockholm, Sweden; xiaoyuan.ren@ki.se; 3Hubei Key Laboratory of Tumor Microenvironment and Immunotherapy, College of Basic Medical Sciences, China Three Gorges University, Yichang 443002, China; wangjfox@gmail.com

**Keywords:** ginger magnolia officinalis, thioredoxin, reactive oxygen species, redox regulation

## Abstract

Some traditional Chinese medicines (TCMs) possess various redox-regulation properties, but whether the redox regulation contributes to antibacterial mechanisms is not known. Here, ginger juice processed Magnoliae officinalis cortex (GMOC) was found to show strong antibacterial activities against some Gram-positive bacteria, but not Gram-negative bacteria including *E. coli*, while the redox-related transcription factor *oxyR* deficient *E. coli* mutant was sensitive to GMOC. In addition, GMOC and its main ingredients, magnolol and honokiol, exhibited inhibitory effects on the bacterial thioredoxin (Trx) system, a major thiol-dependent disulfide reductase system in bacteria. The effects of magnolol and honokiol on cellular redox homeostasis were further verified by elevation of the intracellular ROS levels. The therapeutic efficacies of GMOC, magnolol and honokiol were further verified in *S. aureus*-caused mild and acute peritonitis mouse models. Treatments with GMOC, magnolol and honokiol significantly reduced the bacterial load, and effectively protected the mice from *S. aureus*-caused peritonitis infections. Meanwhile, magnolol and honokiol produced synergistic effects when used in combination with several classic antibiotics. These results strongly suggest that some TCMs may exert their therapeutic effects via targeting the bacterial thiol-dependent redox system.

## 1. Introduction

Traditional Chinese medicine (TCM) has a long history in Chinese pharmacopoeia, with various properties including anticancer, antioxidant, anti-inflammation and so on [1,2,3,4,5]. Although some of the natural extracts have been found to possess antibacterial effects [6,7,8], the action mechanisms were rarely elucidated [9,10,11,12]. Targeting the biofilm [9], peptide deformylase [13] and the cell membrane [14] are the limited reported mechanisms to the best of our knowledge.

The cellular targets of commonly used antibiotics include the bacterial cell wall, bacterial membrane, DNA synthesis, protein synthesis and various others. Recently, several types of antibiotics have been found to stimulate the reactive oxygen species (ROS) production, which contributes to their bactericidal mechanism [15,16,17,18,19]. Therefore, the bacterial thiol-dependent system contributes to mediating the cellular ROS level which emerges as a potential antibacterial target [20,21,22]. There are two major cellular thiol-dependent redox systems, the thioredoxin (Trx) system and the glutathione (GSH) system, which maintain cellular redox homeostasis and protect the cells from oxidative stress [23,24,25,26]. The Trx system is composed of Trx, thioredoxin reductase (TrxR) and nicotinamide adenine dinucleotide phosphate (NADPH), while glutaredoxin (Grx), glutathione reductase, GSH and NADPH form the GSH/Grx system. These two systems have important roles in DNA synthesis and repair, cell proliferation and antioxidant defenses [23,27]. In addition, ferredoxin-NADP^+^ reductase (FNR) can transfer electrons from ferredoxin or flavodoxin molecules to NADP+, participating in the generation of NADPH [28]. The Trx system is involved in a wide range of cellular functions by providing electrons to various critical cellular proteins, including ribonucleotide reductase and methionine-*S*-sulfoxide reductase [29,30,31]. The Trx and GSH-Grx systems have partially overlapping functions and can support each other in mammalian cells [32,33]. The Trx system is widespread in eukaryotes and prokaryotes, but the GSH system is not present in the majority of Gram-positive bacteria [20,34]; thus, bacteria that lack the GSH system are dependent on the Trx system, and the bacterial redox balance is more easily interrupted by inhibition of the Trx system. Interestingly, a ferredoxin-dependent flavin thioredoxin reductase is present in cyanobacteria and clostridia [35], in which Trx acts as pivotar between flavin-oxidizing and flavin-reducing conformations during catalysis [36].

In this paper, a TCM library including more than 300 agents was investigated to repurpose the antibacterial activity, and the GMOC was highlighted. As the two main substances responsible for the beneficial properties of the GMOC [37], magnolol and honokiol were studied in relation to their antibacterial activities in vitro and in vivo. Magnolol and honokiol were previously reported to possess antioxidant, anti-inflammatory and antibacterial pharmacological effects. They have been traditionally used in Chinese and Japanese medicines for the treatment of anxiety, asthma, depression, gastrointestinal disorders and headache [38,39,40,41]. Although many signaling pathways including NF-*κ*B/MAPK, Nrf2/HO-1 and PI3K/Akt pathways are implicated in the mammalian cell biological functions mediated by magnolol and honokiol [40,42]. Their effects of targeting bacterial thiol-dependent redox-related pathways, in particular, the Trx system was beyond investigation. Thus, the inhibitory effects of magnolol and honokiol on Trx and TrxR activities in Gram-positive bacteria such as *S. aureus*, and the destructive effects on the intracellular redox environment were probed in this paper. In addition, their protective capacities over reducing the bacteria load and rescuing mice from *S. aureus*-caused peritonitis infections were also determined. All in all, this study suggests Trx and TrxR as relatively new targets to screen antibacterial agents from the TCM library in the treatment of bacterial infection.

## 2. Materials and Methods

### 2.1. Bacteria Strains and Reagents

TCM formula granules were kindly provided as gifts by Sichuan Neo-Green Pharmaceutical Technology Development Co., Ltd., Chengdu, China. Magnolol and honokiol were purchased from Adamas, Switzerland. Kunming mice were purchased from Chongqing Medical University, Chongqing, China. *E. coli* TrxR and Trx proteins and *E. coli* DHB4 WT and Δ*oxyR* strains were kindly provided by Prof. Arne Holmgren (Karolinska Institutet, Stockholm, Sweden) [14]. MRSA USA300 JE2, *S. aureus* ATCC29213, *S. epidermidis* ATCC14990, *B. subtilis* ATCC6633, *B. cereus* ATCC14579 were from the American Type Culture Collection, USA.

### 2.2. Detection of the Minimal Inhibitory Concentration (MIC) Value of the TCM

Bacteria were cultured in Luria–Bertani (LB) medium at 37 °C, 150 rpm to an OD_600_ of 0.4 and diluted 1: 1000 in the medium for the MIC measurement. Then, 300 different TCM agents with a two-fold serial dilution were cultured with the bacteria in 96-well plates at 37 °C for 24 h, and the OD_600_ was measured using a VERSA micro-well plate reader. The experiments were performed in triplicates to detect the MIC values.

### 2.3. Preparation of Bacterial Cell Lysates

The bacterial cell lysates were extracted from the *S. aureus* inhibited by the antimicrobial agents [21]. Initially, when the *S. aureus* was cultured (37 °C, 220 rpm) until an OD_600_ of 0.4, various concentrations of antimicrobial agents were added, and the culture continued for 3 h at 37 °C to obtain the optimum concentrations of antimicrobial agents to inhibit the bacterial growth. The OD_600_ was measured every 0.5 h and values were calibrated by subtracting the original OD_600_ with the antimicrobial agents. Subsequently, the growth inhibited cultures were diluted 100 times and spread on LB agar plates to determine whether the effects were bactericidal.

To prepare the cell lysate, the bacteria with an OD_600_ of 0.4 were incubated with various concentrations of antimicrobial agents for 10 min. Then, the bacterial cells were harvested by centrifugation (4 °C, 5000 rpm for 5 min). The cells were washed twice with PBS and resuspended in Tris-EDTA buffer (pH 7.4) and protease inhibitor was added to inactivate the protease activity. Finally, the cells were disrupted with a sonicator. Cell lysates were obtained by centrifugation (13,000 rpm for 10 min) and then subjected to further biochemical analysis. The protein concentration was measured by Bradford assay.

### 2.4. Detection of Inhibition of the Bacterial Trx System

The TrxR activity assay was performed in 96-well plates. Cell lysate (25 μg) obtained from the bacteria treated with GMO, magnolol and honokiol was incubated with 2 mM EDTA and 200 μM NADPH at 37 °C for 5 min, then 5 μM *E. coli* Trx and 2 mM DTNB were added [20]. The absorbance at 412 nm was detected using a VERSA microwell plate reader for 5 min, and the slope was used to represent TrxR activity. The activity of Trx from the cell lysate obtained from the bacteria treated with GMO, magnolol and honokiol was determined in 96-well plates [20]. Cell lysate (25 μg) was incubated with 2 mM EDTA and 200 μM NADPH at 37 °C for 5 min, then 100 nM *E. coli* TrxR and 2 mM DTNB were added. The absorbance at 412 nm was detected using a VERSA microwell plate reader for 5 min, and the slope was used to represent Trx activity. The activity of the untreated bacterial sample was used as the control.

### 2.5. Nuclear Staining with Propidium Iodide

Nuclear staining reagent propidium iodide (PI) was used for further research on the inhibition efficiency [21]. Bacteria with an OD_600_ of 0.4 were treated with different concentrations of antimicrobial agents for 10 min at 37 °C. Cells were washed twice with PBS and cells were collected by centrifugation (6000 rpm and 5 min). Nuclei were stained using 5 μg/mL PI for 40 min at 37 °C. Cells were washed twice and resuspended in PBS after the incubation, and the fluorescence measured by flow cytometry (BD FACS Verse).

### 2.6. Inhibition of Bacterial TrxR In Vitro

The DTNB reduction method was used to detect the inhibition of bacterial TrxR activity [20]. In 96-microwell plates, 100 μM NADPH and 50 nM *E. coli* TrxR were added and incubated with different concentrations of antimicrobial agents at room temperature. Another 2.5 μM *E. coli* Trx and 1 mM DTNB were added after incubation; Tris-EDTA (pH 7.4) was used as the control. The absorbance at 412 nm was measured using a VERSA microwell plate reader for 5 min, and the slope was used to represent TrxR activity.

### 2.7. Determination of ROS Levels

Bacteria with an OD_600_ of 0.4 were incubated with antimicrobial agents for 10 min. Then, the bacterial cells were obtained by centrifuging (4 °C, 5000 rpm for 5 min). The pellets were washed three times with PBS and stained with 10 μM H_2_DCFH-DA for 30 min at 37 °C. After the incubation, cells were washed twice and re-suspended in PBS, and the ROS production was quantified by flow cytometry (BD FACS Verse, NJ, USA). For the measurement of HO·, the cells were washed with PBS, then strained with 5 μM HPF for 40 min at 37 °C. After the incubation, cells were washed three times and re-suspended in PBS, and HO· production was quantified by flow cytometry.

### 2.8. Detection of Bacterial Morphology by Electron Microscopies

The experiments were performed as previously described with some modifications [43]. Bacteria were cultured in LB medium until an OD_600_ of 0.4 was reached, and then treated with the antimicrobial agents for 10 min. Cells were obtained by centrifugation (4 °C, 13,000 rpm, 15 min) and fixed with 2.5% glutaraldehyde. The morphology and structure of *S. aureus* cells were observed using scanning electron microscopy with an acceleration voltage of 3 kV (Hitachi SU8020; Hitachi, Ltd., Tokyo, Japan) and transmission electron microscopy (JEM 1200EX; JEOL, Ltd., Tokyo, Japan).

### 2.9. Mild and Acute Peritonitis Mice Model Assay

The mild and acute peritonitis mice experiments were performed using a method modified from previous reports [21,43]. All animal experiments were approved by the Experimental Animal Care & Welfare Committee of Southwest University and abided by the relevant ethical guidelines. Healthy 6-week-old Kunming mice (body weight, 20 ± 2 g) were used for the study. Animals were adaptively housed for 1 week in a specific pathogen-free animal house and had free access to food and water. Animals were fasted for 6–12 h prior to the experiment but were able to drink water. In the mild peritonitis mice model assay, 80 mice were randomly divided into four groups (20 mice/group, half male and half female). All mice were injected intraperitoneally with 200 μL 1.95 × 10^8^ CFU/mL *S. aureus* per mouse to establish a mild peritonitis mice model. One group was orally administered physiological saline as a control. The other groups were orally administered GMOC (5 g/kg body weight), magnolol solution (50 mg/kg body weight) and honokiol solution (50 mg/kg body weight). At 0, 12, 24, 36 and 48 h post-administration, 4 mice per group (half male and half female) were sacrificed, then 1 mL of sterile saline was injected intraperitoneally and the abdomen was massaged. The abdomen was then opened and 200 μL of ascites were collected and serially diluted for the analysis of *S. aureus* CFU/mL.

For the acute peritonitis mice model assay, 40 mice were randomly divided into four groups (10 mice/group, half male and half female). All mice were intraperitoneally injected with *S. aureus* to construct an acute mice peritonitis model. The four groups of mice were administered with physiological saline, GMOC (5 g/kg body weight), magnolol (50 mg/kg body weight) and honokiol (50 mg/kg body weight). Specifically, mice were intraperitoneally administrated with 200 μL 3.9 × 10^8^ CFU/mL *S. aureus* to establish the acute peritonitis mice model. Mice received a single oral administration and were observed for 7 days.

### 2.10. Synergism Measurement

Synergism was measured using the Bliss independent model, as described in a previous study [44], and the synergist degree ‘S’ was calculated using the following formula:

S = (f_X0_/f_00_)(f_0Y_/f_00_) − (f_XY_/f_00_)


f_XY_ refers to the growth rate of bacteria under the two drugs (X and Y) in combination treatment; f_X0_ and f_0Y_ refers to the growth rate of bacteria with one drug treatment only; f_00_ refers to the growth rate of bacteria without drug; S refers to the degree of synergy.

The value of the combined degree S is between −1 and 1. A value closer to −1 indicates an antagonistic effect, and a value closer to 1 indicates a synergistic effect.

## 3. Results

### 3.1. GMOC Was Screened from a TCM Library as the Most Efficient Antibacterial Agent

The MIC values of more than 300 TCM agents were examined by visible spectrophotometry assay in 96-well plates. Among them, 12 TCM agents showed antibacterial capacity against Gram-positive bacteria, including *Staphylococcus aureus* ATCC29213, *Staphylococcus epidermidis* ATCC14990, *Bacillus subtilis* ATCC6633 and *Bacillus cereus* ATCC14579 (Appendix A). Very interestingly, GMOC was found to be the most effective agent with a minimum inhibitory concentration (MIC) of 780.0 μg/mL against all Gram-positive bacteria. All the tested agents did not exhibit antibacterial effects on Gram-negative bacteria *E. coli.* However, the *E. coli* strain with the deletion of *oxyR* gene was sensitive to some of the TCM agents including GMOC. The *oxyR* is a key transcription factor gene involved in various cellular activity mediation including thiol-dependent redox regulation in *E. coli*. Thus, this result indicated that the thiol-dependent redox regulation is a major player in these TCM-induced bacterial growth inhibition.

To confirm the antibacterial activity of GMOC, *S. aureus* ATCC29213 with an OD_600_ of 0.4 were treated with serial concentrations of GMOC. Compared with the control, the growth of *S. aureus* was inhibited by GMOC in a concentration-dependent manner and was completely inhibited by 5000.0 μg/mL GMOC 3 h post-treatment (Figure 1A). Further, the colony formation assay identified that 5000.0 mg/mL GMOC exhibited a bactericidal effect.

To explore whether GMOC affects bacterial Trx system, the TrxR and Trx activities in the cell extracts from *S. aureus* treated with GMOC for 10 min were measured by a 5,5′-dithio-bis-(2-nitrobenzoic acid) (DTNB) reduction assay. The results showed that 1000.0 and 5000.0 μg/mL GMOC could trigger a strong inhibition over both *S. aureus* TrxR activities (Figure 1B).

### 3.2. Magnolol and Honokiol Are the Main Active Ingredients in GMOC Exhibiting Antibacterial Effects

It is well-known that magnolol and honokiol are the main active ingredients in GMOC [45,46,47], thus, antibacterial effects were detected accordingly. *S. aureus*, *S. epidermidis*, *B. subtilis* and *B. cereus* were used in visible spectrophotometry assays, and magnolol and honokiol possessed actively bactericidal activities against four tested Gram-positive bacteria with an MBC around 8.0 μg/mL, but not against the Gram-negative species. Very interestingly, MRSA USA300 JE2 was also sensitive to both magnolol and honokiol (Appendix A). Further, *S. aureus* cells in 15 mL tubes with the OD_600_ of 0.4 were treated with serial concentrations of magnolol and honokiol, respectively. The growth of *S. aureus* was almost completely inhibited by 26.6 μg/mL magnolol (Figure 2A) or 13.3 μg/mL honokiol (Figure 2B), respectively, in a concentration-dependent bactericidal manner.

The propidium iodide (PI) nuclear staining, which represents the bacterial membrane permeability, was performed after treating *S. aureus* cells with serial concentrations of magnolol and honokiol, respectively. PI stains the nucleic acids inside dead cells, or those with damaged membranes. In agreement with the inhibitory effect on the bacterial growth curve, when *S. aureus* was treated with 26.6 μg/mL magnolol or 26.6 μg/mL honokiol, there was nearly 100% of PI-positive cells, indicating a significant rise in bacterial cell death and membrane permeability (Figure 3).

### 3.3. Magnolol and Honokiol Change the Morphology of S. aureus

The effects of 26.6 μg/mL magnolol and 26.6 μg/mL honokiol on the morphology of *S. aureus* cells were detected by electron microscopy. Scanning electron microscopy revealed that the morphology of the *S. aureus* cell membrane changed significantly when treated with magnolol or honokiol compared to the control. Normal *S. aureus* have a smooth surface and a full appearance. After 10 min treatment with magnolol or honokiol, the bacteria were rough and shrunken (Figure 4A). Further, transmission electron microscopy showed that untreated *S. aureus* had a smooth surface and a complete cell membrane and cell wall. After 10 min treatment with magnolol or honokiol, the *S. aureus* cell membrane and cell wall were ruptured, cytoplasmic material flowed out, and the cells eventually died (Figure 4B).

### 3.4. Magnolol and Honokiol Disrupt the Intracellular Redox Micro-Environment

Purified recombinant bacterial TrxR was incubated with serial concentrations of magnolol and honokiol in vitro, and the TrxR activity was determined by DTNB assay. The results showed that the TrxR activity was significantly reduced by 6.7 μg/mL magnolol (Figure 5A) or 6.7 μg/mL honokiol (Figure 5B), respectively, in concentration- and time-dependent manners (Figure 5A,B). The effects of magnolol and honokiol on intracellular TrxR and Trx activities were also detected. Magnolol and honokiol displayed similar inhibitory effects on both the TrxR and Trx activities in *S. aureus.* An amount of 6.7 μg/mL magnolol (Figure 5C) or honokiol (Figure 5D) caused a significant decrease of the activities of TrxR (approximately 70%) and Trx (60%).

The inhibition of the Trx system may induce the thiol-dependent redox micro-environment change, therefore the reactive oxygen species (ROS) levels were determined by flow cytometry. As shown in Figure 6A,B, treatment with magnolol and honokiol on *S. aureus* resulted in a concentration-dependent increase in mean fluorescence intensity (MFI). Furthermore, the *S. aureus* cells were stained with 3′-(p-hydroxyphenyl fluorescein) (HPF) to quantify hydroxyl radical (HO·) production specifically. Treatment with magnolol (above 26.6 μg/mL Figure 6C) or honokiol (above 13.3 μg/mL Figure 6D) induced a corresponding upregulation of the HO· level, respectively. All the above results indicated that magnolol and honokiol could disrupt intracellular redox homeostasis via the inhibition of the Trx system and subsequent ROS elevation.

### 3.5. Therapeutic Efficacy of GMOC in Mice Staphylococcal Peritonitis Infections

To investigate whether the inhibitory effects on the bacteria of GMOC, magnolol and honokiol occur in vivo, two *S. aureus*-induced peritonitis infection mice models were used to examine the therapeutic efficacy. In the experiment with a mild peritonitis infectious model, eighty mice were randomly divided into four groups: control, GMOC, magnolol and honokiol, and were intraperitoneally injected with 200 μL 1.95 × 10^8^ CFU/mL *S. aureus* per mouse. Each group was orally administered an equal volume of normal saline, 5000.0 mg/kg body weight GMOC, 50.0 mg/kg body weight magnolol or 50.0 mg/kg body weight honokiol, respectively. The bacterial load was further measured at 48 h post-administration, and the results showed that the *S. aureus* colony forming units (CFUs)/mL in the peritoneal cavity were decreased in all groups. Although there was no difference among the GMOC, magnolol, and honokiol groups, the CFUs in all three agent-treated groups were significantly decreased compared to the control group (Figure 7A, *p* < 0.001).

In the other acute peritonitis infection experiment, forty mice were randomly divided into four groups: control, GMOC, magnolol and honokiol, and were intraperitoneally injected with 200 μL 3.9 × 10^8^ CFU/mL *S. aureus*. Each group was orally administered an equal volume of normal saline, 5000.0 mg/kg body weight GMOC, 50.0 mg/kg body weight magnolol or 50.0 mg/kg body weight honokiol, respectively. The survival rate was calculated 7 days post-administration, and the results showed that the overall survival was significantly increased in all three agent-treated groups compared to the control group (Figure 7B, *p* < 0.001). The honokiol-treated group had the highest protection (60%), followed by GMOC-treated group (40%), magnolol-treated group (30%) and the control acute peritonitis infection group (0%) (Figure 7B), indicating that honokiol possessed the critical protective activity against *S. aureus*-induced acute peritonitis infection. In all, the above results demonstrated that GMOC, magnolol and honokiol have significant therapeutic efficacies in an *S. aureus*-caused mouse peritonitis model.

### 3.6. Antibacterial Effects of the Combination of Magnolol, Honokiol with Classic Antibiotics against E. coli

Magnolol and honokiol exhibited effective bactericidal effects by disrupting the Trx system, which is an essential system in Gram-positive bacteria; however, magnolol and honokiol failed to kill most Gram-negative bacteria, which have both the Trx and GSH/Grx redox systems. To evaluate the effects of magnolol and honokiol on the antibacterial activity of other classical antibiotics, *E. coli* were treated with the combination of magnolol or honokiol and seven classes of antibiotics. The presence of magnolol or honokiol did not affect the MIC values of erythromycin, streptomycin, amoxicillin, ampicillin, chloramphenicol and ofloxacin. However, when magnolol and honokiol were used in combination with kanamycin, gentamicin and tetracycline, the MIC values of each antibiotic against *E. coli* were significantly lower than when the antibiotic was used alone. Among them, when gentamicin was used alone, the MIC value was 6.3 μg/mL, and when used in combination with magnolol or honokiol, the MIC value decreased to 1.6 μg/mL. When tetracycline was used alone, the MIC value was 3.1 μg/mL, and the MIC values were reduced to 0.8 μg/mL when used in combination with magnolol or honokiol. Thus, magnolol and honokiol exhibited synergistic antibacterial effects in combination with kanamycin, gentamicin or tetracycline.

The synergist degree ‘S’ of the combination of magnolol or honokiol with kanamycin, gentamicin or tetracycline was determined by the Bliss independent model when *E. coli* were treated for 1 and 3 h (Figure 8A). The synergistic effect of the treatment for 1 h was better than the treatment for 3 h. Moreover, the growth curve for *E. coli* was analyzed following the treatment with antibacterial agents alone or in combination. When certain concentrations of magnolol and honokiol were used to treat *E. coli* with these three antibiotics there were significant synergistic effects (Figure 8B). These synergistic effects between magnolol or honokiol, and kanamycin, gentamicin or tetracycline were verified in a PI staining experiment (Appendix A).

To explore the morphological structure changes in *E. coli* after treatment with magnolol and honokiol in combination with gentamicin, the treated bacterial cells were observed with transmission and field scanning electron microscopy. There was a marked change in the morphology of *E. coli* following treatment with the combination of magnolol, honokiol and gentamicin (Figure 8C). In the control group, the *E. coli* cells were smooth, and the cell membrane and cell wall were intact (Figure 8D). Treatment with magnolol, honokiol and gentamicin alone caused a certain degree of damage to the cell membrane and cell wall of *E. coli*. There was a marked increase in cell membrane damage and the outflow of intracellular material following the combination treatment. The surface of the *E. coli* cells became rough and shrunken, and vesicles or irregular protrusions appeared on the surface of the cells. Dissolution on the surface was observed, which may have been caused by the destruction of the bacterial cell wall and efflux of the intracellular material.

As magnolol and honokiol were shown to affect the Trx system in *S. aureus*, it was investigated whether the Trx system in *E. coli* was also altered by the treatment combination. Magnolol, honokiol and gentamicin had a slight or no effect on both of the TrxR and Trx activities in *E. coli* when they were used alone. When combining magnolol or honokiol with gentamicin, the TrxR and Trx activities were all significantly reduced (Figure 9A). The influence of magnolol, honokiol and these antibiotics on the ROS levels in *E. coli* was also detected. There was no significant difference in the ROS production and HO· levels in *E. coli* following the treatment with magnolol, honokiol and the antibiotics alone; only the combination of magnolol or honokiol with gentamicin resulted in a significant elevation of MFI of ROS and HO· (Figure 9B,C).

## 4. Discussion

The opportunistic pathogen *S. aureus* is a leading cause of hospital and immunity-acquired infections, resulting in a broad spectrum of pathology ranging from common skin infections to deep fatal disease [48]. *S. aureus* have evolved resistance to nearly all common classes of antibiotics including β-lactams, aminoglycosides, macrolides, tetracyclines and quinolones [49]. The decreasing effectiveness of conventional drugs is continuously haunting both clinicians and drug researchers, and new targets with a novel strategy against *S. aureus* are urgently needed [50].

Plant natural resources have been demonstrated to possess great chemical and biological diversities and promising findings of antibacterial phytochemicals [51]. In this study, GMOC was screened to be an outstanding antibacterial agent from a TCM agents’ library. Its two main phenolic constituents, magnolol and honokiol, also showed remarkable antimicrobial activities against *S. aureus*. Although their antibacterial activities were reported as early as six decades ago [39,52,53,54], their potential inhibitory activity on the Trx system has not been studied.

Bacterial TrxR is a new potent antibacterial drug target [20,55,56,57,58]. In this study, visible spectrophotometer, transmission electron microscopy, scanning electron microscopy and flow cytometry were used to verify the bactericidal activities of GMOC, magnolol and honokiol against *S. aureus*. Further, all three tested agents showed significant inhibitory effects on the bacterial TrxR and Trx activities. The *S. aureus* Trx system supplies electrons to peroxiredoxins for ROS removal, which is critical for various cellular antioxidants reduction. Thus, inhibition of the Trx system is highly related to the excessive production of ROS (Figure 10).

It was also demonstrated that magnolol and honokiol have potent antibacterial effects against Gram-positive bacteria, which was associated with inhibition of the Trx system and disruption of the intracellular redox environment within the bacteria. The effects of magnolol and honokiol alone on Gram-negative, GSH-positive bacteria were not as potent. This may be due to the partially overlapping functions of the Trx and GSH/Grx systems, which can compensate for each other [59,60,61]. Mammalian cells have both Trx and GSH/Grx systems, whereas most Gram-positive bacteria have no GSH/Grx system and are dependent on the Trx system. Our results, thus, confirmed the Trx system as an antibacterial target, determined the antibacterial mechanism of GMOC, magnolol and honokiol, and analyzed the differences in the antibacterial spectra of GSH-positive and -negative bacteria.

ROS are important molecules involved in many physiological processes; however, excessive production of ROS may cause cell damage or even death of organisms. Studies have shown that the increased production of ROS caused by some clinically-used antibiotics contributes to their bactericidal efficacy [18,62]. Here, *S. aureus* cells stained with H_2_DCFH-DA showed that ROS elevation is one of the major players in determining the bacterial fate, which disrupted the intracellular redox micro-environment. In particular, the HO· level may be a key factor that promotes the cell death.

Further, the *S. aureus*-caused peritonitis mice model was used to determine the therapeutic efficacies of GMOC, magnolol and honokiol on the bacterial burdens and protection abilities. GMOC, magnolol and honokiol showed significant therapeutic efficacies against *S. aureus*-caused peritonitis by significantly reducing the bacterial burden by day 2 (*p* < 0.001); meanwhile, they also highly protected the mice from peritonitis infection, especially magnolol.

## 5. Conclusions

Our findings demonstrated that the active ingredients of GMOC, magnolol and honokiol are promising antibacterial agents that may exert ROS-mediated bactericidal effects by inhibiting the bacterial Trx system. The antioxidant systems within bacteria are major factors that determine the sensitivity of bacteria to these compounds. Our findings may lay a foundation for a novel strategy of antibiotic development by targeting the bacterial Trx system and inducing ROS production.

## Figures and Tables

**Figure 1 antioxidants-12-01180-f001:**
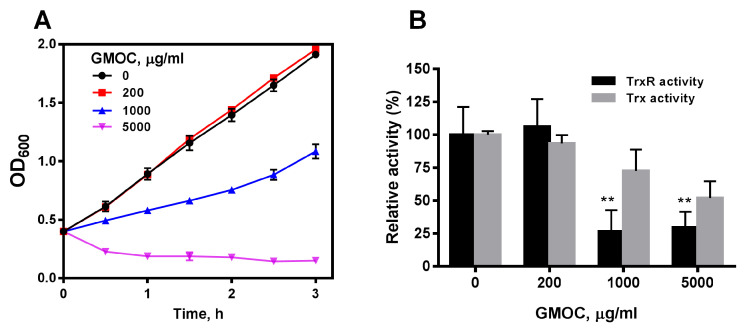
GMOC was screened from a TCM library as the most outstanding antibacterial agent. *S. aureus* ATCC25923. (**A**) Inhibitory effect of GMOC on the growth of *S. aureus*. (**B**) Inhibitory effect of GMOC on *S. aureus* Trx and TrxR activities. Trx (grey) and TrxR (black) activities in the bacterial lysates were assayed using DTNB reduction method. Data are presented as mean ± SD from two experiments. ** *p* < 0.01 (vs. untreated bacteria); two-way ANOVA.

**Figure 2 antioxidants-12-01180-f002:**
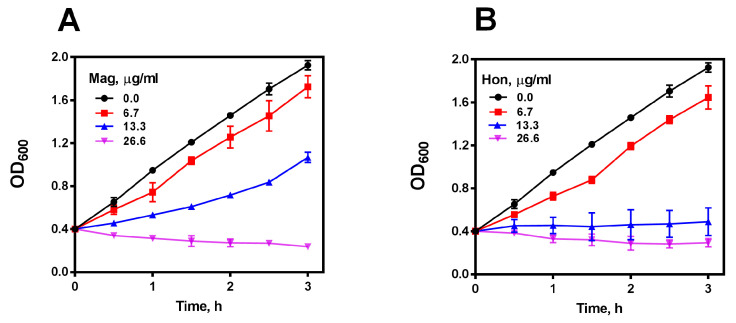
Antibacterial effect of magnolol (Mag) and honokiol (Hon). (**A**) Inhibitory effect of magnolol on the growth of *S. aureus*. (**B**) Inhibitory effect of honokiol on the growth of *S. aureus*. Data are presented as mean ± SD from two experiments.

**Figure 3 antioxidants-12-01180-f003:**
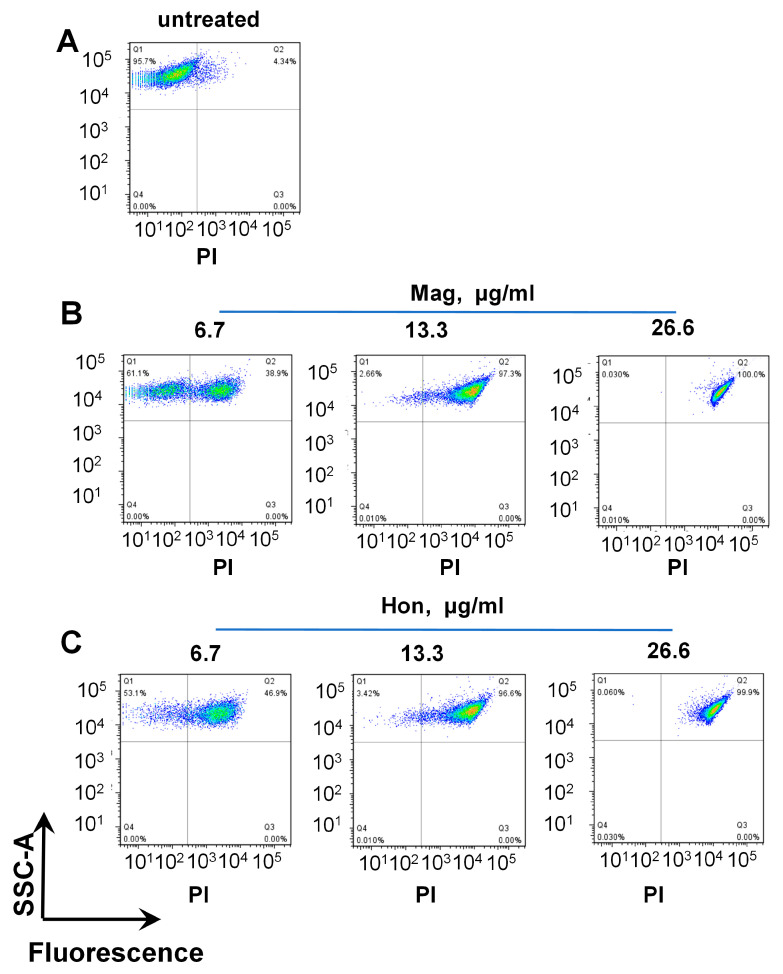
FACS analysis of propidium iodide (PI)-stained *S. aureus*. The *S. aureus* in logarithmic phase was treated with serial concentrations of magnolol (Mag) or honokiol (Hon) for 10 min. The bacteria were stained with PI and analyzed by FACS. (**A**) Untreated bacteria; (**B**) Bacteria treated with magnolol; (**C**) Bacteria treated with honokiol.

**Figure 4 antioxidants-12-01180-f004:**
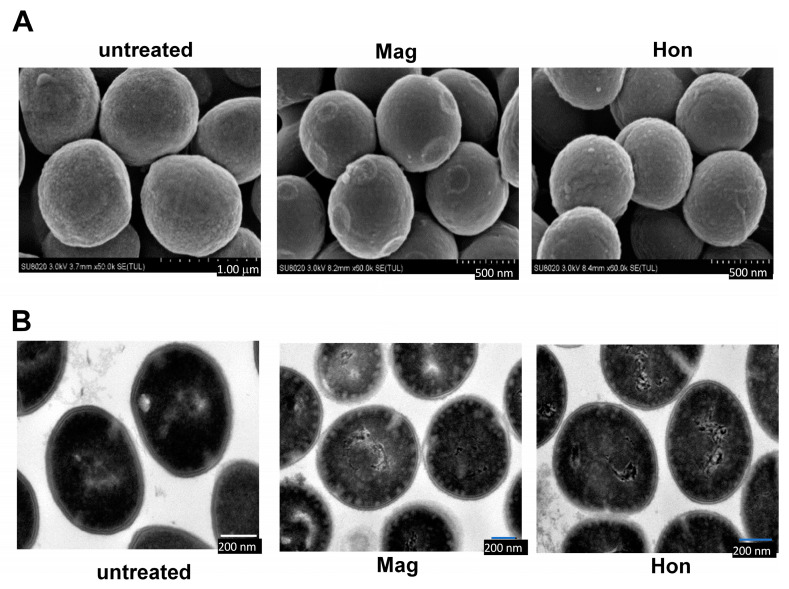
The morphology change of *S. aureus* caused by magnolol and honokiol. *S. aureus* ATCC25923 were treated with the magnolol and honokiol for 10 min. Then the bacteria were fixed with 2.5% glutaraldehyde and detected by electron microscopies. (**A**) Field scanning electron microscopy (Magnification: 60,000×). (**B**) Transmission electron microscopy (Magnification: 12,000×).

**Figure 5 antioxidants-12-01180-f005:**
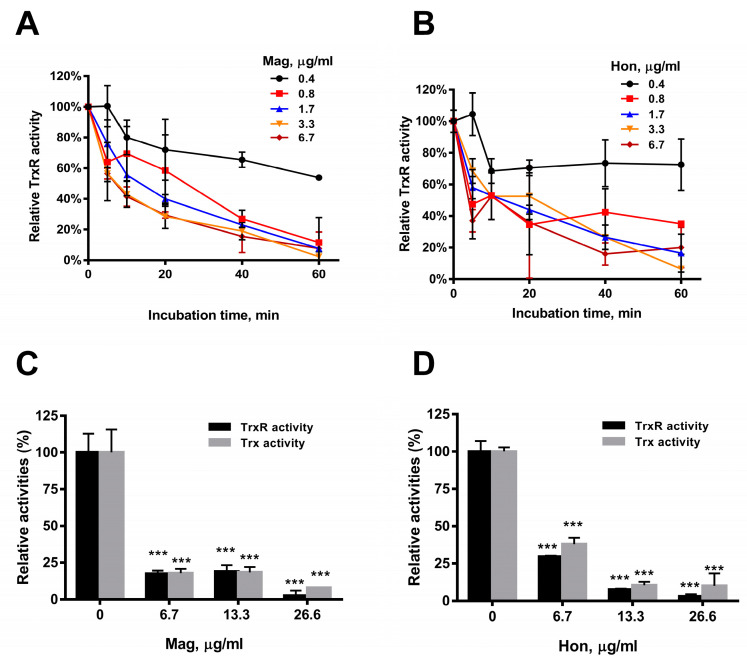
Inhibition of Trx system by magnolol (Mag) and honokiol (Hon). (**A**) Inhibition of purified *E. coli* TrxR activity by magnolol in vitro. TrxR activity was assayed by DTNB reduction method in the presence of purified recombinant Trx in vitro. (**B**) Inhibition of purified *E. coli* TrxR activity by honokiol in vitro. (**C**) Inhibitory effects of the Trx (gray) and TrxR (black) activities in *S. aureus* treated with magnolol for 10 min. (**D**) Inhibitory effects of Trx (gray) and TrxR (black) activities in *S. aureus* treated with honokiol for 10 min. Data are presented as mean ± SD from two experiments. *** *p* < 0.001; two-way ANOVA.

**Figure 6 antioxidants-12-01180-f006:**
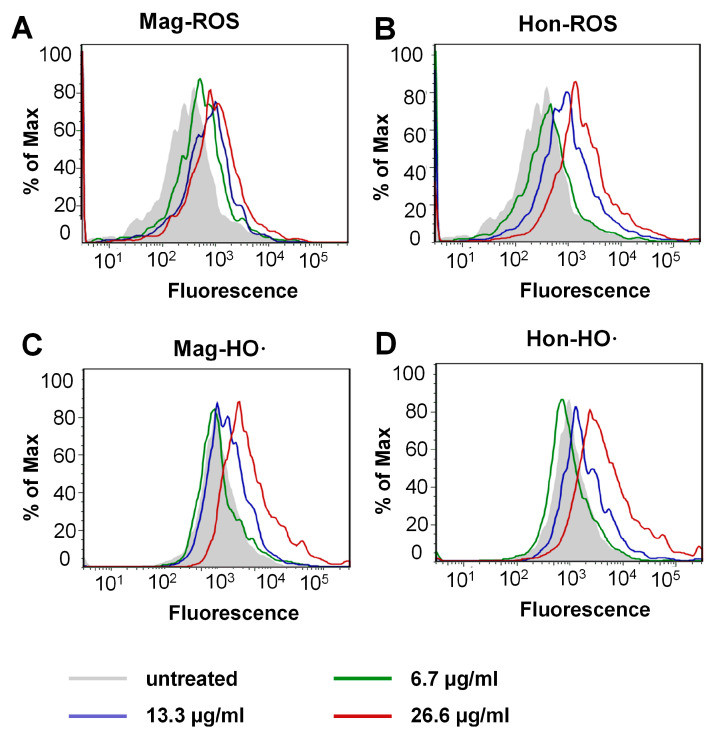
ROS elevation caused by magnolol (Mag) and honokiol (Hon) in *S. aureus*. (**A**,**B**) ROS levels of the bacteria treated with magnolol or honokiol. ROS levels were detected with H_2_DCFH-DA as a probe. (**C**,**D**) HO· levels of the bacteria treated with magnolol or honokiol. HO· levels were detected with HPF.

**Figure 7 antioxidants-12-01180-f007:**
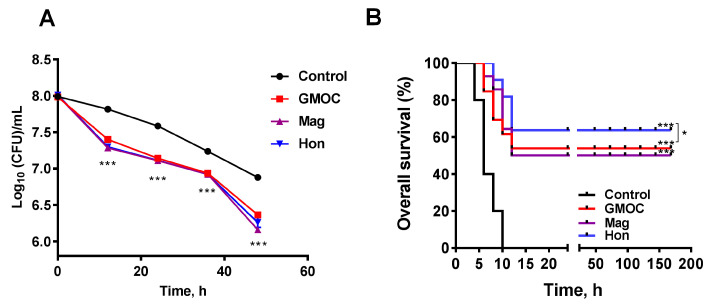
Therapeutic efficacy of GMOC, magnolol and honokiol in *S. aureus*-induced peritonitis mice. (**A**) The effects of 5000.0 mg/kg GMOC, 50.0 mg/kg magnolol, 50.0 mg/kg honokiol on the chronic peritonitis mice infected with *S. aureus*. The treatment with saline was used as the control group. The colony forming units (CFUs)/mL of the *S. aureus* in the peritoneal cavity of the mice in different groups were measured by plate counting method. (**B**) Therapeutic efficacy of GMOC, magnolol and honokiol in acute peritonitis mice. Each group was orally given equal volume of 5000.0 mg/kg GMOC, 50.0 mg/kg magnolol, 50.0 mg/kg honokiol and saline, respectively. The data are shown as means ± SD from four independent experiments, each with triplicate analysis. one-way ANOVA, * *p* < 0.05 (Hon group vs GMOC group); *** *p* < 0.001; (vs peritonitis mice group treated with saline).

**Figure 8 antioxidants-12-01180-f008:**
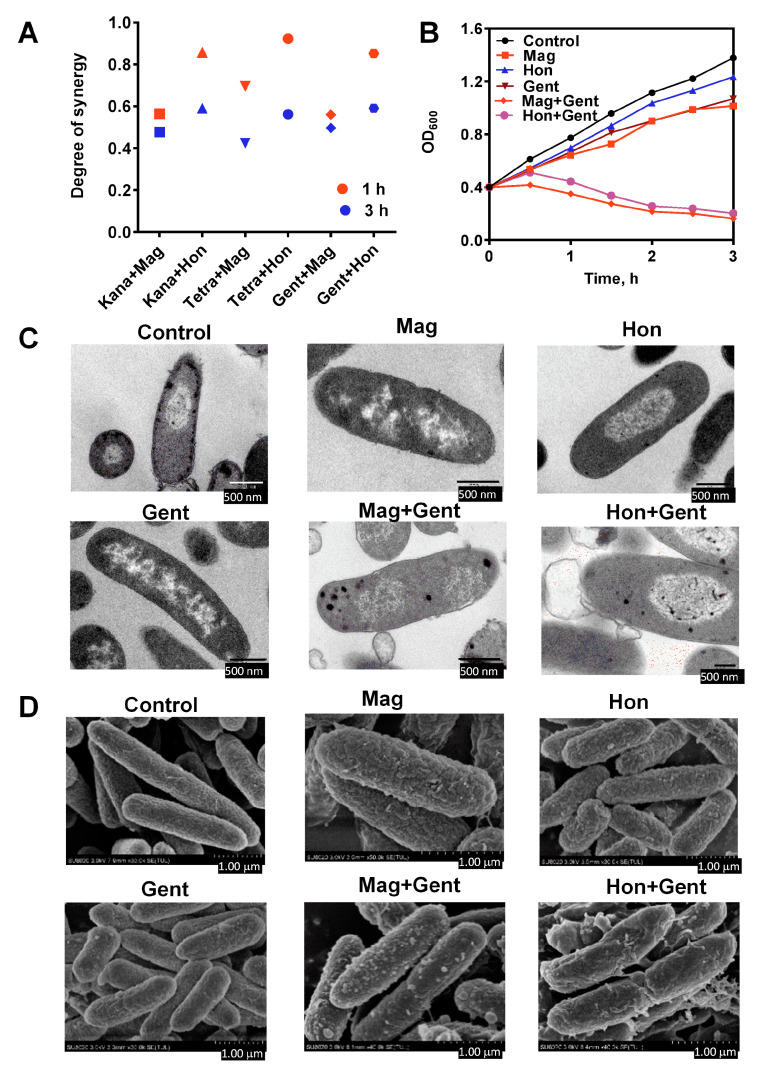
Synergistic effects of the combination of magnolol (Mag) and honokiol (Hon) with kanamycin (Kana), tetracycline (Tetra) and gentamicin (Gent) against *E. coli*. (**A**) Synergistic degree of magnolol and honokiol in combination with antibiotics against *E. coli*. The synergistic degrees were calculated by the Bliss independent model after 1 h (all red) and 3 h (all blue) treatment. (**B**) Effects of the combination of magnolol or honokiol with gentamicin on the growth of *E. coli*. (**C**) Transmission electron microscopy of *E. coli* treated with magnolol and honokiol alone or in combination with gentamicin for 10 min. (Magnification: 60,000×). (**D**) Field scanning electron microscopy of *E. coli* treated with magnolol and honokiol alone in combination with gentamicin. magnolol, 26.6 μg/mL; honokiol, 26.6 μg/mL; kanamycin, 14.6 μg/mL; tetracycline, 12.0 μg/mL; gentamicin, 9.4 μg/mL.

**Figure 9 antioxidants-12-01180-f009:**
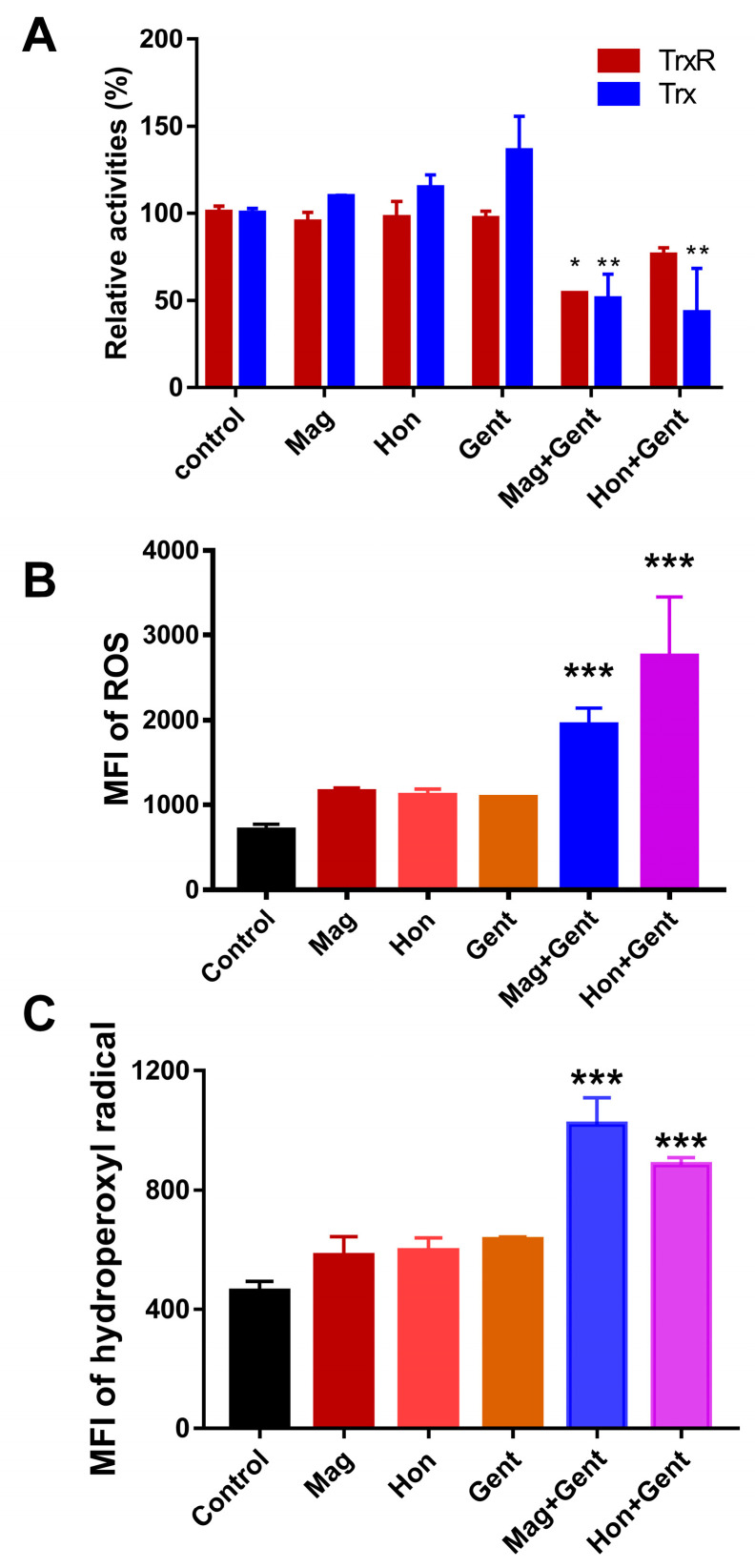
Effects of the combination of magnolol (Mag) and honokiol (Hon) with gentamicin (Gent) on Trx system and ROS level in *E. coli*. (**A**) Effects of the combination of magnolol or honokiol with gentamicin on Trx and TrxR activities in *E. coli*. ***, *p* < 0.001; two-way ANOVA (vs control untreated bacteria) (**B**) Effects of the combination of magnolol or honokiol with gentamicin on the ROS level in *E. coli*. ROS levels were detected with H_2_DCFH-DA as a probe. (**C**) Effects of the combination of magnolol or honokiol with gentamicin on HO· levels in *E. coli*. HO· levels were detected with HPF. 26.6 μg/mL; honokiol, 26.6 μg/mL; gentamicin, 9.4 μg/mL. Data are shown as means ± SD from two or three independent biological experiments. *, *p* < 0.05; **, *p* < 0.01; ***, *p* < 0.001; one-way ANOVA (vs control untreated bacteria).

**Figure 10 antioxidants-12-01180-f010:**
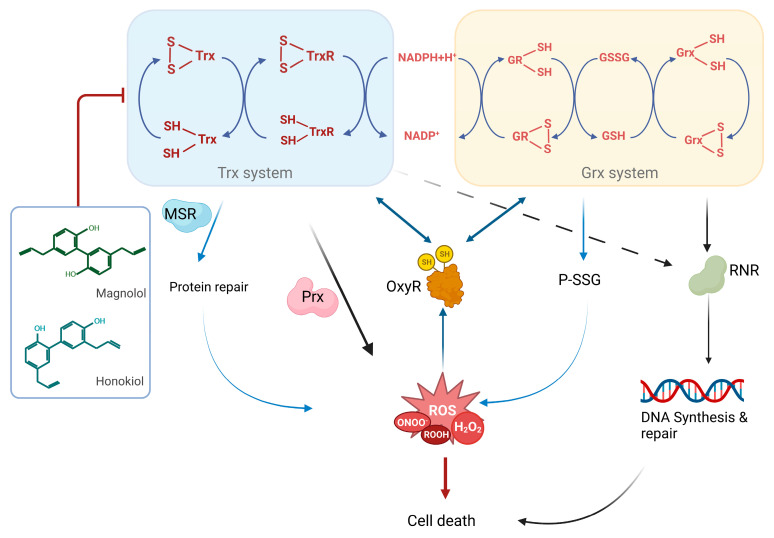
Scheme of the action of magnolol and honokiol in bacteria. The deficiency of GSH system and inactivation of transcription factor OxyR in the Gram-positive bacteria makes the type of bacteria are more sensitive to the active gradients of some traditional Chinese medicine, e.g., magnolol and honokiol. Magnolol and honokiol are excellent inhibitors of the bacterial thioredoxin system. Inhibition of the Trx system results in the elevation of ROS production via the decrease of electron transfer to MSR and Prx. Additionally, the DNA synthesis and repair are also interrupted due to the blockage of electron transfer from the Trx system to RNR, an essential enzyme for the DNA synthesis.

## Data Availability

All of the data is contained within the article and the Appendix A.

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
