# Peer review of "Disruption of Bacterial Thiol-Dependent Redox Homeostasis by Magnolol and Honokiol as an Antibacterial Strategy"

_antioxidants, 2023, doi:10.3390/antiox12061180_

Round 1
Reviewer 1 Report
The manuscript titled “Disruption of Bacterial Thiol-dependent Redox Homeostasis by Magnolol and Honokiol as an Antibacterial Strategy” by Ouyang, Y.; et al. is an original scientific work where the authors study the main ingredients of ginger magnolia officinalis (honokiol and magnolol) and how they affect the redox bacterial homeostasis and the subsequent reactive oxygen species (ROS) production. The authors used many complementary techniques like enzymatic activity assays, flow citometry, scanning electron microscopy, or the employment of animal models to interrogate the effect of honokiol and magnolol on ROS production and how this process affects to Staphylococcus aureus bacteria survival. However, it exists some points that need to be addressed (please, see them below detailed point-by-point). The most relevant outcomes found by the author can contribute to in the growth of many fields like healthcare, development of drug compounds, biomolecular detection and screening or life sciences, among others.
Here, there exists some suggestions in order to improve the scientific quality of the manuscript paper:
1) ABSTRACT. “(…) bacterial Trx system” (line 20). Please, the authors should define thioredoxin. Then, the abbreviation “Trx” should be placed between brackets.
2) INTRODUCTION section is clear and concise. “There are two major cellular thiol-dependent redox systems, the thioredoxin (Trx) system and glutathione (GSH) system, which maintain cellular redox homeostatis and protect cells from oxidative stress” (lines 43-45). Here, the authors should not forget the role of methionine/ methionine sulfoxide (MetO) balance regulated by methionine sulfoxide reductase bacterial enzymes [1,2].
[1] Singh, V.K.; et al. The Role of Methionine Sulfoxide Reductases in Oxidative Stress Tolerance and Virulence of Staphylococcus aureus and Other Bacteria. Antioxidants 2018, 7, 128. https://doi.org/10.3390/antiox7100128.
[2] Boschi-Muller, S. Molecuar Mechanisms of the Methionine Sulfoxide Reductase System from Neisseria meningitidis. Antioxidants 2018, 7, 131. https://doi.org/10.3390/antiox7100131.
3) “The Trx system is composed of Trx, thioredoxin reductase (TrxR) and nicotinamide adenine dinucleotide phosphate (NADPH)” (lines 45-47). Please, the authors should highlight the important role of the C-terminal tyrosine residue in the association between this enzyme:coenzyme system [3].
[3] Pérez-Domínguez, S.; et al. Nanomechanical Study of Enzyme: Coenzyme Complexes: Bipartite Sites in Plastidic Ferredoxin-NADP+ Reductase for the Interaction with NADP+. Antioxidants 2022 11, 537. https://doi.org/10.3390/antiox11030537.
4) “The Trx system is composed (…) NADPH form the GSH/Grx system” (lines 45-48). Here, the manuscript will benefit if the authors add a schematic representation of the chemical reactions taken place for the discussed redox systems.
5) “The Trx system is involved (…) including ribonucleotide reductase, methionide-S-sulfoxide reductase” (lines 49-51). Please, the authors should also indicate the presence of Trx in ferredoxin-dependent flavin thioredoxin reductase (FFTR) enzyme and how Trx acts as pivotar between flavin-oxidizing and flavin-reducing conformations during catalysis [4].
[4] Marcuello, C.; et al. Atomic Force Microscopy to Elicit Conformational Transitions of Ferredoxin-Dependent Flaving Thioredoxin Reductases. Antioxidants 2021, 10, 1437. https://doi.org/10.3390/antiox10091437.
6) MATERIALS AND METHODS. “2.2. Detection of the MIC value of the TCM” (line 85). Plese, the authors should define the abbreviation “MIC” by adding the term “minimal inhibitory concetration”. Then, the authors should specify the supplier name and country of the following technique: “Versa micro-well plate reader” (line 88). Please, take this latest comment into account for the rest of M&M section (e.g. where did the authors purchase the chemical reagents and consumables used in this scientific work?).
7) “The morphology and structure of S. aureus cells were observed usinig scanning electron microscopy” (lines 146-147) What was the acceleration voltage used by the authors? Then, did the authors use any contrast agent? These information details should be reported in this statement.
8) RESULTS. “(…) 0.78 mg/ml. (…) 5.0 mg/ml” (lines 191-200). Please, the authors should homogenize the significant figures. This point should be taken into account for the rest of the main manuscript body text.
9) Figure 1 (line 209). The lettering of the figure axes and the respective caption is not legible. Maybe this problem comes during the pdf conversion. The authors should check this point in order to improve the readability for the potential target stakeholders. This comment is extandable for the rest of the Figures.
10) Figure 4 (line 255). The authors should indicate better the respective scale bars displayed in the different pannels. Same comment for Figure 8 (line 357).
11) “The effects of magnolol and honokiol on intracellular TrxR and Trx activities (…) 6.66 µg/ml magnolol (Figure 5C) and 6.66 µg/ml honokiol (…) in S. aureus” (lines 267-270). Does this sentence mean that magnolol and honokiol display the same inhibitiory effects? Please, the authors should add a brief statement in this regard.
12) DISCUSSION AND CONCLUSIONS sections. The authors perfectly remarks the most significant outcomes found in this work. No actions are requestes for these sections.
13) REFERENCES. The references are in the proper format style of Antioxidants. No actions are requested in this section.
------------------------------------
OVERVIEW AND FINAL COMMENTS
The submitted work is well-designed and the gathered results are interesting to have a more complete outlook of the ROS-mediated role that honokiol and magnolol play against bacterial viability. This knowledge could significantly aid to design the next-generation of antimicrobial compounds. For these reasons, I will recommend the present scientific manuscript for further publication in Antioxidants once all the aforementioned suggestions will be properly fixed.
The scientific paper is well written in general terms. Neverthelss, the authors should recheck the manuscript in order to improve final details.
Author Response
Reviewer 1:
The manuscript titled “Disruption of Bacterial Thiol-dependent Redox Homeostasis by Magnolol and Honokiol as an Antibacterial Strategy” by Ouyang, Y.; et al. is an original scientific work where the authors study the main ingredients of ginger magnolia officinalis (honokiol and magnolol) and how they affect the redox bacterial homeostasis and the subsequent reactive oxygen species (ROS) production. The authors used many complementary techniques like enzymatic activity assays, flow cytometry, scanning electron microscopy, or the employment of animal models to interrogate the effect of honokiol and magnolol on ROS production and how this process affects to Staphylococcus aureus bacteria survival. However, it exists some points that need to be addressed (please, see them below detailed point-by-point). The most relevant outcomes found by the author can contribute to in the growth of many fields like healthcare, development of drug compounds, biomolecular detection and screening or life sciences, among others.
>>>Many thanks for these comments!
Here, there exists some suggestions in order to improve the scientific quality of the manuscript paper:
1) ABSTRACT. “(…) bacterial Trx system” (line 20). Please, the authors should define thioredoxin. Then, the abbreviation “Trx” should be placed between brackets.
>>>The thioredoxin has been defined as “thioredoxin (Trx)”.
2) INTRODUCTION section is clear and concise. “There are two major cellular thiol-dependent redox systems, the thioredoxin (Trx) system and glutathione (GSH) system, which maintain cellular redox homeostatis and protect cells from oxidative stress” (lines 43-45). Here, the authors should not forget the role of methionine/ methionine sulfoxide (MetO) balance regulated by methionine sulfoxide reductase bacterial enzymes [1,2].
[1] Singh, V.K.; et al. The Role of Methionine Sulfoxide Reductases in Oxidative Stress Tolerance and Virulence of Staphylococcus aureus and Other Bacteria. Antioxidants 2018, 7, 128. https://doi.org/10.3390/antiox7100128.
[2] Boschi-Muller, S. Molecuar Mechanisms of the Methionine Sulfoxide Reductase System from Neisseria meningitidis. Antioxidants 2018, 7, 131. https://doi.org/10.3390/antiox7100131.
>>>Thank the reviewer to point out the two papers. The methionine/ methionine sulfoxide (MetO) balance regulated by methionine sulfoxide reductase-mediated oxidation and reduction of methionine residues are also an important antioxidant event in which thioredoxin systems are involved. We have added the two papers in the manuscript as references (line 52).
3) “The Trx system is composed of Trx, thioredoxin reductase (TrxR) and nicotinamide adenine dinucleotide phosphate (NADPH)” (lines 45-47). Please, the authors should highlight the important role of the C-terminal tyrosine residue in the association between this enzyme: coenzyme system [3].
[3] Pérez-Domínguez, S.; et al. Nanomechanical Study of Enzyme: Coenzyme Complexes: Bipartite Sites in Plastidic Ferredoxin-NADP+ Reductase for the Interaction with NADP+. Antioxidants 2022 11, 537. https://doi.org/10.3390/antiox11030537.
>>> The sentence about NADPH generation by ferredoxin-NADP+ reductase (FNR) and related reference has been added.
4) “The Trx system is composed (…) NADPH form the GSH/Grx system” (lines 45-48). Here, the manuscript will benefit if the authors add a schematic representation of the chemical reactions taken place for the discussed redox systems.
>>> Thanks for the suggestion. We have added a scheme figure as the figure 10.
5) “The Trx system is involved (…) including ribonucleotide reductase, methionide-S-sulfoxide reductase” (lines 49-51). Please, the authors should also indicate the presence of Trx in ferredoxin-dependent flavin thioredoxin reductase (FFTR) enzyme and how Trx acts as pivotar between flavin-oxidizing and flavin-reducing conformations during catalysis [4].
[4] Marcuello, C.; et al. Atomic Force Microscopy to Elicit Conformational Transitions of Ferredoxin-Dependent Flaving Thioredoxin Reductases. Antioxidants 2021, 10, 1437. https://doi.org/10.3390/antiox10091437.
>>> Fdx-dependent flavin TRs (FFTRs) is one interesting type of TrxR which has been described in cyanobacteria and clostridia, we have added the related paper in the manuscript.
6) MATERIALS AND METHODS. “2.2. Detection of the MIC value of the TCM” (line 85). Please, the authors should define the abbreviation “MIC” by adding the term “minimal inhibitory concentration”. Then, the authors should specify the supplier’s name and country of the following technique: “Versa micro-well plate reader” (line 88). Please, take this latest comment into account for the rest of M&M section (e.g. where did the authors purchase the chemical reagents and consumables used in this scientific work?).
>>> The definition of minimal inhibitory concentration (MIC) has been added. The companies and the suppliers of the materials and equipment have been included.
7) “The morphology and structure of S. aureus cells were observed using scanning electron microscopy” (lines 146-147) What was the acceleration voltage used by the authors? Then, did the authors use any contrast agent? These information details should be reported in this statement.
>>> The parameter of the analysis has been added.
8) RESULTS. “(…) 0.78 mg/ml. (…) 5.0 mg/ml” (lines 191-200). Please, the authors should homogenize the significant figures. This point should be taken into account for the rest of the main manuscript body text.
>>> Thank the reviewer for this suggestion. The significant figures have been homogenized in the whole manuscript.
9) Figure 1 (line 209). The lettering of the figure axes and the respective caption is not legible. Maybe this problem comes during the pdf conversion. The authors should check this point in order to improve the readability for the potential target stakeholders. This comment is extandable for the rest of the Figures.
>>> Thank the reviewer for this suggestion. The resolution and fonts of figures have been improved.
10) Figure 4 (line 255). The authors should indicate better the respective scale bars displayed in the different pannels. Same comment for Figure 8 (line 357).
>>> Thank the reviewer for this suggestion. The resolution and fonts of figures have been improved.
11) “The effects of magnolol and honokiol on intracellular TrxR and Trx activities (…) 6.66 µg/ml magnolol (Figure 5C) and 6.66 µg/ml honokiol (…) in S. aureus” (lines 267-270). Does this sentence mean that magnolol and honokiol display the same inhibitory effects? Please, the authors should add a brief statement in this regard.
>>> Thank the reviewer for this suggestion. A sentence to describe the inhibitory effects of magnolol and honokiol has been added.
12) DISCUSSION AND CONCLUSIONS sections. The authors perfectly remark the most significant outcomes found in this work. No actions are requests for these sections.
>>>Thanks for your affirmation.
13) REFERENCES. The references are in the proper format style of Antioxidants. No actions are requested in this section.
>>>Thanks for your affirmation.
Reviewer 2 Report
In the manuscript entitled »Disruption of Bacterial Thiol-dependent Redox Homeostasis by Magnolol and Honokiol as an Antibacterial Strategy« the antibacterial properties of Ginger magnolia officinalis and its two main phenolic constituents, magnolol and honokiol, against the opportunistic pathogen S. aureus was investigated. The authors show that all three compounds have potent antibacterial effects against S. aureus associated with inhibition of the bacterial Trx system and disruption of the intracellular redox environment within the bacteria. The authors suggest that the Trx system may be a novel target for antibacterial drug development and that stimulating the production of ROS may be a promising strategy for bacterial eradication.
Overall, the study presents interesting results. The manuscript is well written, however the following errors need to be corrected.
Throughout the manuscript:
Since GMO stands for genetically modified organisms, I suggest using a different abbreviation.
Standardisation of units throughout the manuscript, as the MIC is once given in mg/mL and another time in µg/mL.
Introduction
Line 35: The authors should add the following citations:
* Molecules 2020, 25(12), 2947
* Plants 2020, 9(12), 1680
Materials and Methods
2.2 What was the final concentration of the bacteria?
Why is the highest value and not the mean value reported?
How was the MIC value determined? Visually?
Which growth medium was used?
2.3 Written very vaguely. Was growth checked only during the first three hours or during the entire 24 hours? So were the MIC values determined after 3 or 24 hours?
2.4 With what were the cells washed?
It should be indicated which samples were examined in each experiment.
Results
3.1 Line 191 MIC against which bacteria?
Line 198 Change serial concentrations to serially diluted concentrations.
How can the MIC value of GMO be 0.78 mg/mL, if bacterial growth was not completely inhibited until 5 mg/mL?
Figure fonts or even entire figures should be larger because the labels are difficult to read.
OK.
Author Response
Reviewer 2:
In the manuscript entitled »Disruption of Bacterial Thiol-dependent Redox Homeostasis by Magnolol and Honokiol as an Antibacterial Strategy« the antibacterial properties of Ginger magnolia officinalis and its two main phenolic constituents, magnolol and honokiol, against the opportunistic pathogen S. aureus was investigated. The authors show that all three compounds have potent antibacterial effects against S. aureus associated with inhibition of the bacterial Trx system and disruption of the intracellular redox environment within the bacteria. The authors suggest that the Trx system may be a novel target for antibacterial drug development and that stimulating the production of ROS may be a promising strategy for bacterial eradication.
Overall, the study presents interesting results. The manuscript is well written, however the following errors need to be corrected.
Throughout the manuscript:
Since GMO stands for genetically modified organisms, I suggest using a different abbreviation.
>>>According to reference PMID: 35318753, GMO has been changed to GMOC, ginger juice processed Magnoliae officinalis cortex.
Standardization of units throughout the manuscript, as the MIC is once given in mg/mL and another time in µg/mL.
>>> The unit throughout the manuscript has been standardized as µg/mL.
Introduction
Line 35: The authors should add the following citations: * Molecules 2020, 25(12), 2947* Plants 2020, 9(12), 1680
>>> The two papers have been added.
Materials and Methods
2.2 What was the final concentration of the bacteria? Why is the highest value and not the mean value reported? How was the MIC value determined? Visually? Which growth medium was used?
>>>The theoretical final concentration of the bacteria OD600=0.0004. Since the limit of detection of the microplate reader is OD600=0.04, then we chose to directly present the procedure. To determine MIC, repeated experiments have been performed. We presented the highest result to show that all other obtained results lower than the presented value. To avoid confusion, “the highest” has been deleted. MIC value was determined by UV-vis assay, which presented 90% bacterial inhibition. Bacteria were cultured in Luria-Bertani (LB) medium.
2.3 Written very vaguely. Was growth checked only during the first three hours or during the entire 24 hours? So were the MIC values determined after 3 or 24 hours?
>>> Thank the reviewer for pointing out this description. The MIC values were determined after entire 24 h incubation. The culture for 3h was used to determine optimum concentration of antibacterial agents to inhibit the bacterial growth with OD600=0.4, so that we obtained enough sample for the preparation of cell extract and further biochemical analysis. We have rewritten this part to make it more clearly.
2.4 With what were the cells washed? It should be indicated which samples were examined in each experiment.
>>>The bacterial cells washed twice using PBS. The samples for the analysis were indicated.
Results
3.1 Line 191 MIC against which bacteria? Line 198 Change serial concentrations to serially diluted concentrations. How can the MIC value of GMO be 0.78 mg/mL, if bacterial growth was not completely inhibited until 5 mg/mL?
>>> The bacteria which were inhibited by GMOC were inserted in the sentence. MIC value of GMOC is 0.78 mg/mL, which was measure with bacteria OD600=0.0004, while in the experiment to detect the inhibition of bacterial growth, the bacterial OD600 was 0.4. The initial bacterial concentration difference may cause drug concentration difference.
Figure fonts or even entire figures should be larger because the labels are difficult to read.
>>> The figures’ resolution and fonts have been improved.
Reviewer 3 Report
The manuscript entitled "Disruption of Bacterial Thiol-dependent Redox Homeostasis by Magnolol and Honokiol as an Antibacterial Strategy" by Ouyang and co-authors describe the antimicrobial effects of traditional Chinese Medicines (TCMs), in particular Ginger magnolia officinalis (GMO) and their components magnolol and honokiol. The authors found that the antimicrobial activity is more effective towards gram-positive bacteria, in particular Staphylococcus aureus. Furthermore, the susbstances investigated showed a relatively ineffective towards gram-negative bacteria, in particular Escherichia coli. The authors further investigate the effects of these compounds on the Trx and TrxR systems, upon which rely the antioxidant activities of gram-positive bateria. Although this is not ompletely new, the authors brought some evidence on the molecular basis of the atimicrobial activity of TCMs and in particular Ginger magnolia officinalis (GMO) and their components magnolol and honokiol.
The manuscript is well organized and written. There are however some experiments that were performed in conditions that are not according to the standards and the results are therefore difficult to consider. Namely the MIC values and the inhibitory effects of compounds studied were determined after 3 hours of incubation. The addition of inhibitory compounds to microbial populations often result in an adaptation phase which could last for hours before the normal growth or inhibited growth restarts. The authors should therefore determine the inhibitory effect of compounds after 24 h and not 3 hours. this is in my view the major drawback of the study and needs to be considered.
Other issues:
lines 51-64: the sentence is too long and dificult to understand, Rephrase as follows: "Magnolol and honokiol previously reported as classic multifunctional neolignans have been traditionally used in Chinese and Japanese medicines for the treatment of anxiety, asthma, depression, gastrointestinal disorders and headache, with notably antioxidant, anti-inflammatory, antibacterial and pharmacological effects".
line 73: The Trx and TrxR systems are not novel targets, they are already known. Rephrase.
lines 86-89: No information on the culture conditions or culture conditions is provided. Further details are required.
line 104: "protein inhibitor was added to inactivate the protease activity". protease ihibitor, not protein inhibitor.
lines 111-114: "The activity of Trx was determined with same procedure [18] except instead of 5 μM E. coli Trx with 100 nM E. coli TrxR in the reaction mixture." The sentence makes no sense, rewrite.
line 116: the abbreviation PI needs to be explained.
lines 194-196: The sentence "The oxyR is a key transcription factor gene involved in thiol-dependent redox regulation in E. coli." The sentence is incorrect, as the oxyR regulates more general functions.
line 196: ... in these...
lines 200-201: "Further, the colony formation assay identified that 5.0 mg/ml GMO exhibited a bactericidal effect." A better description is required. What is the experimental basis used to lead to this conclusion?
The YY and XX axis labels of figures are very dificcult to read, increase the letter size.
Figure 1: the xx label, fluorescence, is presented in which units?
The English usage requires minor revisions.
Author Response
Reviewer 3:
The manuscript entitled "Disruption of Bacterial Thiol-dependent Redox Homeostasis by Magnolol and Honokiol as an Antibacterial Strategy" by Ouyang and co-authors describe the antimicrobial effects of traditional Chinese Medicines (TCMs), in particular Ginger magnolia officinalis (GMO) and their components magnolol and honokiol. The authors found that the antimicrobial activity is more effective towards gram-positive bacteria, in particular Staphylococcus aureus. Furthermore, the substances investigated showed a relatively ineffective towards gram-negative bacteria, in particular Escherichia coli. The authors further investigate the effects of these compounds on the Trx and TrxR systems, upon which rely the antioxidant activities of gram-positive bacteria. Although this is not completely new, the authors brought some evidence on the molecular basis of the antimicrobial activity of TCMs and in particular Ginger magnolia officinalis (GMO) and their components magnolol and honokiol.
The manuscript is well organized and written. There are however some experiments that were performed in conditions that are not according to the standards and the results are therefore difficult to consider. Namely the MIC values and the inhibitory effects of compounds studied were determined after 3 hours of incubation. The addition of inhibitory compounds to microbial populations often result in an adaptation phase which could last for hours before the normal growth or inhibited growth restarts. The authors should therefore determine the inhibitory effect of compounds after 24 h and not 3 hours. this is in my view the major drawback of the study and needs to be considered.
>>> Thank the reviewer for pointing out this description. The MIC values were determined after entire 24 h incubation. The culture for 3h was used to determine optimum concentration of antibacterial agents to inhibit the bacterial growth with OD600=0.4, so that we obtained enough sample for the preparation of cell extract and further biochemical analysis. We have rewritten this part to make it more clearly.
Other issues:
lines 51-64: the sentence is too long and difficult to understand, Rephrase as follows: "Magnolol and honokiol previously reported as classic multifunctional neolignans have been traditionally used in Chinese and Japanese medicines for the treatment of anxiety, asthma, depression, gastrointestinal disorders and headache, with notably antioxidant, anti-inflammatory, antibacterial and pharmacological effects".
>>>The whole sentence has been revised.
line 73: The Trx and TrxR systems are not novel targets, they are already known. Rephrase.
>>>The sentence has been rephrased to “Trx and TrxR as relatively new targets’.
lines 86-89: No information on the culture conditions or culture conditions is provided. Further details are required.
>>>Bacteria were cultured in Luria-Bertani (LB) medium at 37ËšC 150 rpm to an OD600 of 0.4.
line 104: "protein inhibitor was added to inactivate the protease activity". protease ihibitor, not protein inhibitor.
>>>Protein inhibitor has been changed to protease inhibitor.
lines 111-114: "The activity of Trx was determined with same procedure [18] except instead of 5 μM E. coli Trx with 100 nM E. coli TrxR in the reaction mixture." The sentence makes no sense, rewrite.
>>>Thanks for pointing out this. The whole part was rephrased to “The activity of Trx of cell lysate obtained from the bacteria treated with GMO, magnolol and honokiol was determined in 96-well plates [20]. Cell lysate (25 μg) was incubated with 2 mM EDTA and 200 μM NADPH at 37ËšC for 5 min, then 100 nM E. coli TrxR and 2 mM DTNB were added. The absorbance at 412 nm was detected using a VERSA mi-crowell plate reader for 5 min, and the slope was used to represent Trx activity.”
line 116: the abbreviation PI needs to be explained.
>>>PI stands for Propidium iodide.
lines 194-196: The sentence "The oxyR is a key transcription factor gene involved in thiol-dependent redox regulation in E. coli." The sentence is incorrect, as the oxyR regulates more general functions.
>>> The description of the sentence is changed.
line 196: ... in these...
>>> The misspelling has been corrected.
lines 200-201: "Further, the colony formation assay identified that 5.0 mg/ml GMO exhibited a bactericidal effect." A better description is required. What is the experimental basis used to lead to this conclusion?
>>>After plating the treated bacterial, the colony formation assay identified that 5.0 mg/ml GMO exhibited a bactericidal effect. This was described in 2.3. first paragraph.
The YY and XX axis labels of figures are very difficult to read, increase the letter size.
>>> The figures’ resolution and fonts have been improved.
Figure 1: the xx label, fluorescence, is presented in which units?
>>> Thank the reviewer for this suggestion. It may be Figure 3, we have added PI in the xx label.
Round 2
Reviewer 2 Report
The authors successfully resolved all issues raised by this reviewer. Consequently, the manuscript has been significantly improved and can be in its current version recommended for publication in Antioxidants.
Ok.
Reviewer 3 Report
The criticisms raised were adequately addressed by the authors. The revised version is now more clear and no further issues were identified.